# Body Composition and Nutritional Status of the Spanish National Breaking Team Aspiring to the Paris 2024 Olympic Games

**DOI:** 10.3390/nu15051218

**Published:** 2023-02-28

**Authors:** Cristina Montalbán-Méndez, Nuria Giménez-Blasi, Inés Aurora García-Rodríguez, José Antonio Latorre, Javier Conde-Pipo, Alejandro López-Moro, Miguel Mariscal-Arcas, Nieves Palacios Gil-Antuñano

**Affiliations:** 1Medicine, Endocrinology and Nutrition Service, Sports Medicine Center, General Sports Sub-Directorate, Higher Sports Council (CSD), 28040 Madrid, Spain; 2Nutrition Area, Faculty of Health Sciences, Catholic University of Avila, 05005 Ávila, Spain; 3Department Food Technology, Nutrition and Food Science, Campus of Lorca, University of Murcia, 30100 Murcia, Spain; 4Department of Nutrition and Food Science, School of Pharmacy, University of Granada, 18012 Granada, Spain; 5Instituto de Investigación Biosanitaria (ibs.GRANADA), 18012 Granada, Spain

**Keywords:** body composition, nutritional status, Breaking

## Abstract

Breaking is a sports dance modality that will debut for the first time at the Paris 2024 Olympic Games. This dance form combines street dance steps with acrobatics and athletic elements. It complies with gender equality, maintains aesthetic appeal, and is practised indoors. The objective of this study is to assess the characteristics of body composition and nutritional status of the athletes that make up the Breaking national team. This national team was recruited, and they underwent an analysis of body composition using bioimpedance measurement and a nutritional interview status with the completion of a survey on the frequency of the consumption of sports supplements and ergogenic aids. In addition, they completed a consumption questionnaire for a series of food groups with specified protein, lipid, and carbohydrate content. After that, parameters were analyzed in relation to their nutritional status during a complete medical examination at the Endocrinology and Nutrition Service of the Sports Medicine Center of CSD. A descriptive analysis of the results obtained was carried out to find the mean values of the variables analyzed. The analytical parameters described an adequate nutritional status, except for the mean capillary determination of 25-OH-vitamin D3, which was 24.2 ng/dL (SD: 10.3). Bone mineral density values were higher than those of the general population. This is the first time that a study of these characteristics has been carried out on Breakers, so it is highly relevant to increase knowledge in this area in order to conduct nutritional interventions aimed at improving the sports performance of these athletes.

## 1. Introduction

Breaking is a form of sports dance, whose origin dates back to the 1970s in the streets of the New York neighbourhood of the Bronx [1]. It progressively evolved during the 1980s and gained followers all over the world in the 1990s, becoming part of the International Sports Dance Federation (WSDF). In Spain, the Spanish Federation of Sports Dance (SFSD) was established in 2011, and two years later it was integrated as part of the Spanish Olympic Committee [2]. Given its great acceptance at the 2018 Youth Games, in 2020 Breaking was included in the list of sports that will participate for the first time in the 2024 Paris Olympic Games, being the first sports dance specialty that will participate as an Olympic sport.

Breaking is a dance modality that combines street dance steps with acrobatics and athletic elements. Like the rest of sports dance modalities, Breaking meets the following requirements [2]: (a) proposes gender equality, (b) has great appeal from an aesthetic and adaptability point of view, and (c) be practiced in closed areas.

The level of aerobic capacity measured by maximum oxygen consumption (VO2max) has been studied in Breaking practitioners and has been compared with other high-level sports dance specialties [3,4]. The types of injuries presented by these athletes have also been analyzed compared with other types of modern dance and with other sports specialties [5,6,7], showing that the injuries of these athletes occur more frequently and have a different etiology than other dance modalities. A 2015 study [8] described the biomechanics of Breaking movements, which justify the high level of injuries in the lower limbs, comparing the frequency and intensity of these injuries with sports such as gymnastics.

However, there is little evidence on the anthropometric and body composition characteristics of Breakers. A recent study [9] analyzed the body composition of dancers from four types of urban dance specialties to assess asymmetries in relation to the type of technical movements performed in the corresponding discipline and thus be able to relate it to the most frequent type of injuries that occur. In this study, they concluded that both male and female Breakers have significant asymmetries in the lean mass of both upper limbs. Zaletel [10] qualitatively analyzed the knowledge of nutrition and the use of nutritional supplements in Breaking in relation to other sports dance modalities.

To date, however, no studies have been carried out on the assessment of the nutritional status and body composition of Breaking practitioners, nor has the relationship between these two variables been compared with other sports dance modalities or with other Olympic sports such as gymnastics.

The objective of this work is to examine the characteristics of the body composition and the nutritional status of the national team that aspires to represent Spain in the Olympic Games (JJ) in Paris in 2024. As secondary objectives, we plan to complete the following tasks: (A) analyze the body composition of the Breaking athletes using electrical bioimpedanciometry, (B) know the nutritional status of these athletes, (C) analyze the ergogenic aids used by these athletes in order to improve sports performance, and (D) reflect on possible interventions at a nutritional level to optimize sports performance.

## 2. Material and Methods

This is an analytical, observational, and cross-sectional research study in which the characteristics of body composition and nutritional status of the athletes of the national Breaking team were evaluated.

For this study, 8 members of the national Breaking team (aged between 18 and 35 years old) were recruited. The dancers were concentrated in the area around the High-Performance Center (CAR) of Madrid, Spain and were awarded scholarships to train in this facility, given the inclusion of Breaking in the Olympic Games in Paris in 2024. The team is a representative sample of Breaking dancers in Spain. The study subjects were summoned to the Endocrinology and Nutrition Unit in the Sports Medicine Center of the CAR in Madrid between March and April 2022 to undergo a complete medical examination. In addition, an informed consent was offered to all the participants that informed them about the objectives of the study and asked for their voluntary participation. The informed consent was designed following the information model of the Spanish Ministry of Health [11].

As the first part of the examination, a fasting blood test was taken, and the following blood parameters were analyzed:

Biochemical: creatinine (mg/dL), urea (mg/dL), glucose (mg/dL), lipid profile (total cholesterol and triglycerides in mg/dL), and micronutrients (serum iron in µg/dL, ferritin in ng/dL, calcium, phosphorus and magnesium in mg/dL, potassium in mmol/L, and vitamin B12).

Blood count: hemoglobin (g/L), hematocrit (%), and total leukocytes and platelets (n°/mm^3^). Second, an anthropometric study was performed, measuring height (cm) using a SECA stadiometer. Body composition analysis was performed by electrical bioimpedance measurement (BIA) with a body composition analyser (InBody 720, Microcaya, Bilbao, Spain), analyzing the following body parameters: weight (kg), body mass index (BMI) (kg/m^2^), skeletal muscle mass (SMM) (kg), % fat, and % body water. Bone mineral densiometry was measured using dual energy X-ray absorptiometry (DEXA), which is the gold standard in research as it is the method with which all other bone measurement methods are compared. This technique uses very low doses of radiation without side effects or prior anaesthesia and is a non-invasive test. It measures bone density and strength. To undergo this test, athletes signed an informed consent form beforehand. It is performed without metallic objects on the body (rings, watches, etc.) and without having undergone any contrast test in the previous two weeks. The athlete is placed in a supine position, stretched out and with arms parallel to the body without opening the fingers; a tape is placed around the ankles to ensure that the legs are immobilised, and the body is checked to ensure that it is within the measurement frame. The test lasts between 6 and 10 min, depending on height and weight. The DEXA sends an invisible thin beam with two energy peaks, which is examined by the machine and sent to the software and represented in a report [12]. For the Breaking athletes, the bone densitometry obtained a lumbar spine T-score and a femoral hip T-score. Subsequently, a complete medical examination was carried out in which capillary vitamin D3 levels were determined (measured in ng/mL).

Finally, a nutritional interview was carried out that included a survey on the frequency of consumption (FFQ) of sports supplements and ergogenic aids. Information was obtained on the Break Dancers types of supplements and dosage patterns. In addition, information on the foods consumed from food groups, such as rice, pasta, or bread; protein from meat, fish, eggs, or nuts; and the consumption of vegetables and fruit was gathered. The information allows us to know the food types the athletes consume on a daily, weekly, or monthly basis. The food consumption frequency questionnaire was previously validated by the research group [12,13].

The rigorous screening procedure allowed us to examine both independent (sex, age, years of sports practice, height) and dependent (weight, BMI, percentage of water, percentage of fat, SMM, analytical levels of nutritional markers) variables.

The data were collected in a database where the different variables analyzed were entered. Non-parametric statistical analysis was performed through descriptive study (maximum, minimum, mean, and SD) and the Spearman correlation test, using SPSS version 22.0 software and following the CHAMP declaration [14].

The researchers followed the ethical principles [15] of respect for the human being, benefit–risk balance, voluntary participation, free and informed consent, respect for the privacy, dignity, and convictions of the participant, and the responsibility and competence of the researchers. This consent was authorized by the Ethics Committee of the UOC University (Spain) and complies with the Helsinki regulations.

## 3. Results

A total of eight subjects were recruited, among whom seven were men and one was a woman (Table 1). The mean age of the sample was 27.5 years old (SD: 5.3) and the mean time spent practicing sports in Breaking was 13.5 years (SD: 4.9).

Regarding anthropometric and body composition characteristics (Table 1), the sample had a mean height of 170.7 cm (SD: 6.6), mean weight of 68.7 kg (SD: 5.3), BMI of 23.7 kg/m^2^ (SD: 1.3), skeletal muscle mass (SMM) of 33.9 kg (SD: 3.9), body fat percentage of 13.0% (SD: 5.5), and body water percentage of 45.3% (SD: 5.4).

In terms of bone mineral density, a mean femoral head T-score of 2.98 (SD: 1.21) and a lumbar spine T-score of 2.20 (SD: 1.42) were obtained. The mean capillary 25-OH-vitamin D3 determination was 24.2 ng/dL (SD: 10.3) (Table 2).

The biochemical parameters analysed (Table 2) showed the following mean values: glucose 88.4 mg/dL (SD: −6.4), creatinine 0.94 mg/dL (SD: 0.1), urea 38.3 mg/dL (SD: 9.7), total cholesterol 168.8 mg/dL (SD: 23.8), triglycerides 64.7 mg/dL (SD: 26.3), total protein 7.23 g/dL (SD: 0.4), potassium 4.58 mmol/L (SD: 0.26), magnesium 1.95 mmol/L (SD: 016), calcium 9.86 mg/dL (SD: 0.266), phosphorus 3.50 mg/dL (SD: 0.31), serum iron 100.82 µg/dL (SD: 42.1), and ferritin 102.5 ng/dL (SD: 63.42). Finally, hemoglobin 14.9 g/dL (SD: 1.1), hematocrit 44.0% (SD: 2.9), leucocytes 5370/mm^3^ (SD: 1741), and platelets 263.25/mm^3^ (SD: 58.89) (Table 2).

The ergogenic aids and sports supplementation used by the study participants are shown in Table 3. Of the eight subjects in the sample, six athletes used some type of ergogenic aid, all of them male. A total of three subjects used creatine in a dose of 3 g before training. Caffeine was used to enhance performance during training by two of the subjects. One of the subjects used an isotonic drink during training, and a total of three team members used whey protein powder supplementation in doses of 20–30 g after training to support recovery. Other aids used to a lesser extent are reflected in Table 3 below.

Table 4 and Table 5 show the correlations between the variables under study. When correlating the body composition variables with biochemical and analytical values and weekly frequency of food intake (Table 4), a positive correlation was observed for the weight of the subjects under study with creatinine (*p* = 0.007) and weekly frequency of pasta consumption (*p* = 0.048); subjects’ height correlated with iron (*p* = 0.015), calcium (*p* < 0.001), hemoglobin (*p* = 0.01), hematocrit (*p* = 0.047), pasta consumption (*p* = 0.043), and meat consumption (*p* = 0.021) and negatively with vegetable consumption (*p* = 0.038). Muscle mass correlated positively with creatinine (*p* < 0.001) and pasta consumption (*p* = 0.038) and negatively with platelets (*p* = 0.015). Fat % correlated negatively with phosphorus (*p* = 0.043), while BMI correlated positively with fruit consumption (*p* = 0.028) and negatively with leucocytes (*p* = 0.04).

When correlating the weekly food consumption of the study population with their biochemical and analytical values (Table 5), total protein was positively correlated with the consumption of pulses and nuts (*p* = 0.031). Urea correlated positively with the consumption of rice (*p* = 0.037) and white fish (*p* = 0.032) and negatively with the consumption of vegetables (*p* = 0.048) and eggs (*p* = 0.040). Magnesium correlated positively (*p* = 0.043) with dairy consumption, phosphorus negatively with bread consumption (*p* = 0.014), ferritin positively with meat consumption (*p* = 0.042), hematocrit with legume consumption (*p* = 0.035), and finally, platelets correlated negatively with rice consumption (*p* = 0.01) and positively with vegetable consumption (*p* = 0.033).

## 4. Discussion

The results obtained in relation to body composition are within the normal range when compared to the general population with normal weight, as the BMI is between 18.5 and 25 kg/m^2^. Body fat percentage is low compared to the general population (around 13% on mean) but in line with other athlete populations [16]. Body water percentage is similar to other studies with similar characteristics, and muscle mass is slightly higher [9]. In the study by Prus Dasa [9], in which anthropometric asymmetries were evaluated based on the study of body composition, it was observed that the group of Breaking athletes (*n* = 22) presented body composition values similar those of this study in terms of BMI, MME, and % water. In contrast, the body fat % of our group of Break Dancers is somewhat higher than the reference study. The mean age of the participants was 20 years old (SD: 2.00), although the mean height was slightly higher than those of the Spanish team, 177.00 cm (SD: 5.60). Our sample of Break Dancers is more heterogeneous in age than that found in comparative studies, as our group has a maximum of 35 years and a minimum of 18 years (SD: 5.34). The limited scientific literature on this sport makes it difficult to find comparative studies that are better adjusted to our age range. The use of different anthropometric formulas in the different studies is another limitation to be taken into account.

The analytical values included in this study are within the normal range, according to the reference values used by the laboratory of the CSD Sports Medicine Center [17], but they cannot be compared with other Breaking athletes because there is no evidence in this regard in the literature. These results show that the nutritional status from the analytical point of view of the analysed athletes is adequate, as they do not present micronutrient deficiencies or metabolic alterations, such as glycemic, lipid, or renal function profile.

In relation to VO2 max for Break Dancers, it was possible to compare them with theatrical dancers, finding that our type of athlete performs exercises of short duration, which allows them a greater cardiorespiratory demand, while theatrical dancers have higher Vo2 max peaks [3].

The mean of the sample under study in relation to bone mineral density was within the normal range considering this entire sample, with the femoral and lumbar spine T-score, both above two. This means that the values of bone mineral density are optimal for these Break Dancers in the competitive period in which they find themselves, far from presenting a risk of fragility and bone injuries that could be related to malnutrition or overtraining. These results support the importance that a non-deficient diet has on the impact of sports performance in this sport discipline.

Mean capillary vitamin D3 levels show mild insufficiency (values between 20 and 30 ng/dL) in our sample of Breaking athletes. These values are very frequent in the winter and spring months and are also more frequent in athletes training in indoor facilities. Normal values for the general population should be above 30 ng/dL, and for the athlete population vitamin D3 levels should be between 20 and 50 ng/dL, depending on the series [18]. Comparisons with other samples of Breaking athletes cannot be made because no series with these results have been published. Athletes with levels in the range of mild insufficiency were prescribed vitamin D3 supplementation (calcifediol 0.266 mg or 15,000 IU), one softgel every 15 days for 6–8 weeks and were scheduled for a repeat measurement and re-evaluation of treatment. In addition, a nutritional intervention was performed to increase intake of vitamin-D-rich oily fish and to ensure sun exposure at least 15 min per day without sunscreen.

The use of sports supplementation by the subjects in the sample is very heterogeneous. The role of ergogenic support is to improve athletic performance through supplementation in conjunction with a varied and balanced diet. More than half of the team uses at least one ergogenic aid with the aim of improving sports performance and/or body composition. In Zaletel’s study [10], which assesses the supplementation of several dance specialities, the results differ greatly from those obtained in this study, as the most used supplements are energy bars and isotonic drinks during training, while the use of multivitamins and protein supplements is similar to the data obtained in this study.

This study is a pioneering study of the body composition and nutritional status of high-level Breaking athletes who have been selected to debut at the next Olympics in Paris 2024. To date, no similar study has been carried out, nor are the normal values of the population studied known.

A quantitative analysis of the intake of the subjects in the sample would have been useful in order to be able to objectively assess whether the energy intake is in line with the estimated energy expenditure. In a review, Benardot analyses the body composition and nutritional status of artistic athletes [19], including dancers and gymnasts, concluding that the tendency of these athletes is to maintain a relative energy deficit, below their requirements, which is associated with poorer results in body composition, lower performance, and greater risk of injury. This energy deficit is often compensated for by an excessive use of sports supplementation, which is why nutritional interventions are necessary in these athletes to avoid erroneous behaviour and ensure an optimal energy balance.

The positive correlation values of creatinine, together with the weight of the athletes, may be related to the generally higher muscle mass in athletes, since total muscle mass is the most important determinant of muscle creatine content and the production derived from its waste, creatinine. This is also supported by the results obtained in the muscle mass values of the subjects under study and their creatinine [20]. It is interesting that the weekly frequency of pasta consumption is correlated with the weight of the subjects and their muscle mass, since it would be directly related to energy needs and consumption, where it seems that this increase in energy needs would be covered by an increase in the consumption of carbohydrate-rich foods such as pasta [21].

Phosphorus is involved in energy production and makes up the most representative energy unit in muscle (ATP and creatine phosphate), so it is directly linked to exercise metabolism. It works together with calcium, and it is ideal to maintain a good balance close to 1:1. Both the % fat of the subjects studied, and the consumption of bread have an inverse correlation with the serum phosphorus of these subjects, although in neither case is it deficient, which could justify the optimization of sports performance in subjects with a low % fat and a high use of energy from anaerobic sources [22].

BMI, in the case of this population of elite athletes is not used as an indicator of overweight/obesity but as another parameter of body composition. It shows a clear relationship with fruit consumption, which indicates that we are dealing with well-advised professionals. The increase in their energy needs, derived from the adaptations inherent in training, such as the increase in muscle mass and the consequent increase in BMI, is covered with carbohydrate source foods, such as fruit, pasta, or bread. We found a negative association with leucocytes as in the study by Ryder et al. [23]. The high biological protein value of plant foods comes mainly from legumes and nuts, and we found an association between serum protein and the frequency of consumption of these two types of food, determining the importance that their consumption could have for the optimal performance of elite athletes.

The urea levels found in the population are high and correlated with the declared frequency of rice and white fish consumption, so it would be advisable to study this fact in depth and make dietary adaptations that would benefit better urea values [24] by controlling the amounts of food to be avoided in uremia. Magnesium seems to benefit from the frequency of consumption of dairy products in this population [25], as does ferritin with meat consumption and hematocrit with consumption of legumes [26], all of which are very important for optimum performance and therefore foods that would favour the health and sporting improvements of these subjects.

This study establishes for the first-time scientific evidence on the body composition and nutritional status of high-level sports performance Breakers. As a new Olympic sport, this discipline must be understood in depth at both a technical and functional level in order to be able to carry out the necessary interventions from the point of view of training and sports nutrition with the aim of optimising the sporting performance of its practitioners. This type of study should be extended to other Breaking teams and other federated sportsmen and women in order to obtain data from a representative sample on the body composition of these sportsmen and women, as is available in other sports specialities. In addition, it could be of interest to extend the study of body composition with other tools, such as plicometry, dual energy X-ray absorptiometry (DEXA), and phase angle by electrical bioimpedance. Regarding knowledge on the nutritional assessment of Breaking athletes, other lines of research to increase knowledge on the subject would be to carry out indirect or direct calorimetry to determine the energy requirements of these athletes, analyse the frequency of consumption of the different food groups, and analyse a greater number of analytical parameters in relation to nutritional status, the level of hydration, and the degree of overtraining.

All this information should be compared with other dance modalities and/or sports modalities of an aesthetic/artistic nature (such as gymnastics or acrobatics) to establish similarities and differences of Breakers with other sports.

We encountered several limitations. Firstly, the sample size is too small to obtain significant conclusions for the study population. It would be necessary to carry out other studies, including a greater number of Breaking athletes to be able to know the anthropometric characteristics, body composition, and nutritional status of high-performance Breakers in order to be able to compare these data with other sports specialities or with the modalities of sports gymnastics.

Furthermore, in this study, both sexes were included in the sample analysed, although there was only one female subject, so the results may be biased by the lack of stratification of the sample. Further studies should be carried out in the future that include a greater number of male and female subjects in order to be able to know the body composition of Breaking athletes in both sexes and thus achieve the objectives of this work.

## 5. Conclusions

The body composition values of the Breaking athletes of the national team that will compete in the next Olympic Games in 2024 are within the normal range for the general active population and are similar to those of previous studies carried out on these athletes. The analytical assessment of the nutritional status is within the normal range, so the Breakers do not show signs of malnutrition or overtraining. The consumption of pulses, meats, dairy products, and nuts seems to influence better biochemical values of parameters related to the sports performance of the subjects under study, so they could be suitable foods for elite and Olympic athletes. The femoral and lumbar bone mineral density values are within the normal range in the sample analyzed and are higher than those of the general population. Capillary vitamin D3 levels show a mild insufficiency, so it is advisable to evaluate these levels in Breaking athletes in order to prescribe adequate supplementation and optimise sporting performance. The use of sports supplementation and/or ergogenic aids in Breakers is highly variable and heterogeneous. Further studies on the body composition and nutritional status of Breakers are needed to obtain reference values for this population and to be able to make appropriate interventions to maximise the sporting performance of these athletes.

## Figures and Tables

**Table 1 nutrients-15-01218-t001:** Description of the study sample and anthropometric and body composition variables analysed in the initial descriptive study of Breaking athletes.

	Minimum	Maximum	Mean	SD
Current age (yrs)	18.00	35.00	27.50	5.34
Years of practice (yrs)	5.00	20.00	13.50	4.95
Height (cm)	158.40	178.80	170.71	6.64
Weight (kg)	61.30	75.50	68.76	5.37
BMI (kg/m^2^)	21.70	25.30	23.70	1.30
SMM (kg)	25.80	38.00	33.95	3.98
FAT (%)	7.50	24.90	13.08	5.55
% Water	34.20	52.50	45.38	5.47

BMI: body mass index; SMM: skeletal muscle mass.

**Table 2 nutrients-15-01218-t002:** Bone densitometry values (T-score), vitamin D3, analytical values (biochemical and hemogram) analysed in Breaking athletes.

	Minimum	Maximum	Mean	SD
Bone mineral density femoral (T-Score)	0.28	3.67	2.08	1.22
Bone mineral density lumbar (T-Score)	0.47	4.65	2.20	1.42
25-OH-Vit D3 (ng/mL)	4.20	34.20	24.25	10.34
Total protein (g/dL)	7.00	8.00	7.23	0.43
Total cholesterol (mg/dL)	145.00	224.00	168.84	23.82
Triglycerides (mg/dL)	40.00	110.00	64.70	26.34
Glucose (mg/dL)	81.00	99.00	88.44	6.43
Urea (mg/dL)	28.00	59.00	38.30	9.70
Creatine (mg/dL)	0.81	1.16	0.94	0.12
Potassium (mmol/L)	4.00	5.00	4.56	0.26
Magnesium (mg/dL)	2.00	2.00	1.96	0.16
Phosphorus (mg/dL)	3.00	4.00	3.50	0.31
Serum iron (µg/dL)	46.00	173.00	100.82	42.10
Ferritin (ng/mL)	24.00	219.00	102.51	63.42
Calcium (mg/dL)	9.00	10.00	9.86	0.26
Hemoglobin (g/dL)	13.00	17.00	14.94	1.11
Hematocrit (%)	40.00	49.00	44.03	2.89
Leukocytes (10^3^/mm^3^)	3.00	8.00	5.37	1.74
Platelets (10^3^/mm^3^)	204.00	382.00	263.25	58.89

**Table 3 nutrients-15-01218-t003:** Sports supplementation and ergogenic aids used by the Spanish National Breaking Team.

	N
**Use of supplementation or ergogenic aids**	
YES	6
NO	2
**Products (ergogenic aids)**	
Whey protein (20–30 g post-training)	3
Isotonic drink	1
Caffeine (variable dose)	2
Creatine (3 g pre-training)	3
BCAA (post-training)	1
Glutamine (pre-training)	1
Beta-alanine (2.5 g post-training)	1
Multivitamin supplement	2

BCAA: branched-chain amino acids.

**Table 4 nutrients-15-01218-t004:** Correlations ^‡^ of body composition variables under study with analytical and biochemical values and frequency of food consumption.

	Weight (Kg)		Height (cm)		SMM ^‡‡^ (Kg)		% Fat		BMI (Kg/m^2^)		% Body Water		BMD ^‡‡‡^ Femoral		BMD ^‡‡‡^ Lumbar	
	R	P	R	P	R	P	R	P	R	P	R	P	R	P	R	P
25-OH-Vit D3 (ng/mL)	0.071	0.867	−0.548	0.120	0.095	0.015	0.214	1.000	0.707	0.180	−0.238	0.693	−0.405	0.651	−0.452	0.289
Total protein (g/dL)	0.095	0.823	0.381	0.352	−0.262	0.531	0.238	0.570	−0.263	0.528	−0.048	0.911	0.357	0.385	0.357	0.385
Total cholesterol (mg/dL)	−0.143	0.736	−0.119	0.779	0.000	1.000	0.143	0.736	0.036	0.933	−0.214	0.610	0.214	0.610	−0.262	0.531
Triglycerides (mg/dL)	−0.467	0.243	−0.323	0.435	−0.084	0.844	−0.551	0.157	−0.325	0.432	0.467	0.243	0.252	0.548	0.036	0.933
Glucose (mg/dL)	0.500	0.207	0.095	0.823	0.429	0.289	−0.071	0.867	0.455	0.257	0.119	0.779	−0.143	0.736	0.000	1.000
Urea (mg/dL)	0.381	0.352	0.595	0.120	0.524	0.183	−0.095	0.823	−0.216	0.608	−0.119	0.779	−0.286	0.493	0.000	1.000
Creatine (mg/dL)	0.855 **	0.007	0.482	0.227	0.952 **	<0.001	−0.434	0.283	0.388	0.342	0.530	0.177	−0.181	0.668	0.518	0.188
Potassium (mmol/L)	−0.181	0.668	0.337	0.414	−0.193	0.647	−0.133	0.754	−0.667	0.071	0.084	0.843	0.120	0.776	0.265	0.526
Magnesium (mg/dL)	0.335	0.417	0.168	0.691	0.132	0.756	0.168	0.691	0.271	0.516	−0.311	0.453	−0.647	0.083	−0.359	0.382
Phosphorus (mg/dL)	−0.265	0.526	−0.289	0.487	0.193	0.647	−0.723 *	0.043	−0.200	0.635	0.482	0.227	−0.422	0.298	−0.084	0.843
Serum iron (µg/dL)	0.571	0.139	0.810 *	0.015	0.333	0.420	0.190	0.651	−0.036	0.933	−0.286	0.493	−0.357	0.385	0.024	0.955
Ferritin (ng/mL)	0.405	0.320	0.310	0.456	0.452	0.260	−0.500	0.207	−0.048	0.910	0.619	0.102	−0.119	0.779	0.643	0.086
Calcium (mg/dL)	0.587	0.126	0.934 **	<0.001	0.503	0.204	−0.096	0.821	−0.235	0.575	0.000	1.000	−0.180	0.670	0.252	0.548
Hemoglobin (g/dL)	0.619	0.102	0.833 *	0.010	0.476	0.233	−0.048	0.911	−0.096	0.821	−0.024	0.955	−0.238	0.570	0.190	0.651
Hematocrit (%)	0.405	0.320	0.714 *	0.047	0.333	0.420	−0.167	0.693	−0.240	0.568	0.167	0.693	0.095	0.823	0.333	0.420
Leukocytes (10^3^/mm^3^)	−0.405	0.320	0.095	0.823	−0.310	0.456	−0.238	0.570	−0.731 *	0.040	0.190	0.651	0.381	0.352	0.190	0.651
Platelets (10^3^/mm^3^)	−0.595	0.120	−0.405	0.320	−0.810 *	0.015	0.333	0.420	−0.180	0.670	−0.167	0.693	0.548	0.160	−0.071	0.867
Frequency of weekly consumption (times a week)																
Pasta	0.711 *	0.048	0.723 *	0.043	0.735 *	0.038	−0.241	0.565	0.091	0.830	0.193	0.647	−0.458	0.254	0.361	0.379
Rice	0.540	0.167	0.135	0.750	0.552	0.156	0.037	0.931	0.383	0.349	0.000	1.000	−0.160	0.706	0.086	0.840
Bread	0.528	0.179	0.466	0.244	0.172	0.684	0.405	0.319	0.327	0.429	−0.233	0.578	0.086	0.840	0.160	0.706
Pulses	0.263	0.528	0.359	0.382	0.108	0.799	−0.192	0.649	−0.102	0.809	0.287	0.490	0.024	0.955	0.359	0.382
Meat	0.589	0.124	0.786 *	0.021	0.454	0.258	−0.172	0.684	−0.179	0.671	0.295	0.479	0.025	0.954	0.663	0.073
White fish	0.426	0.293	0.551	0.157	0.275	0.509	0.275	0.509	−0.101	0.812	−0.225	0.592	0.275	0.509	0.175	0.678
Oily fish	0.263	0.528	0.252	0.548	0.419	0.301	−0.287	0.490	−0.163	0.700	0.419	0.301	0.431	0.286	0.551	0.157
Dairy products	0.037	0.931	0.135	0.750	−0.074	0.862	−0.025	0.954	−0.142	0.737	−0.061	0.885	−0.135	0.750	−0.135	0.750
Vegetables	−0.422	0.298	−0.735 *	0.038	−0.699	0.054	0.518	0.188	0.321	0.438	−0.193	0.647	0.639	0.088	−0.084	0.843
Fruits	0.000	1.000	−0.683	0.062	−0.195	0.643	0.317	0.444	0.761 *	0.028	−0.220	0.601	−0.244	0.560	−0.439	0.276
Eggs	−0.036	0.933	−0.108	0.799	0.024	0.955	−0.443	0.272	0.000	1.00	0.575	0.136	0.144	0.734	0.359	0.382
Nuts	0.000	1.000	0.263	0.528	−0.180	0.670	−0.024	0.955	−0.265	0.526	0.311	0.453	0.611	0.108	0.635	0.091

^‡^ Spearman Rho; ^‡‡^ skeletal muscle mass; ^‡‡‡^ bone mineral density. * *p* ≤ 0.05; ** *p* ≤ 0.001.

**Table 5 nutrients-15-01218-t005:** Correlations ^‡^ of analytical and biochemical values with frequency of food consumption.

	Pasta		Rice		Bread		Pulses		Meat		White Fish		Oily Fish		Dairy Products		Vegetables		Fruits		Eggs		Nuts	
	R	P	R	P	R	P	R	P	R	P	R	P	R	P	R	P	R	P	R	P	R	P	R	P
BMD ^‡‡^ Femoral	−0.458	0.254	−0.160	0.706	0.086	0.840	0.024	0.955	0.025	0.954	0.275	0.509	0.431	0.286	−0.135	0.750	0.639	0.088	−0.244	0.560	0.144	0.734	0.611	0.108
BMD ^‡‡^ Lumbar	0.361	0.379	0.086	0.840	0.160	0.706	0.359	0.382	0.663	0.073	0.175	0.678	0.551	0.157	−0.135	0.750	−0.084	0.843	−0.439	0.276	0.359	0.382	0.635	0.091
25-OH-Vit D3 (ng/mL)	−0.012	0.977	0.479	0.230	−0.160	0.706	−0.671	0.069	−0.565	0.145	−0.225	0.592	−0.108	0.799	−0.479	0.230	0.193	0.647	0.610	0.108	−0.371	0.365	−0.683	0.062
Total protein (g/dL)	0.084	0.843	−0.479	0.230	0.417	0.304	0.755 *	0.031	0.651	0.081	0.100	0.814	−0.204	0.629	0.466	0.244	0.072	0.865	−0.244	0.560	0.335	0.417	0.755*	0.031
Total cholesterol (mg/dL)	−0.325	0.432	0.233	0.578	0.049	0.908	−0.671	0.069	−0.589	0.124	0.350	0.395	0.311	0.453	−0.430	0.288	0.217	0.606	−0.073	0.863	−0.383	0.349	−0.311	0.453
Triglycerides (mg/dL)	−0.612	0.107	−0.309	0.457	−0.408	0.316	0.048	0.910	−0.531	0.176	−0.252	0.547	0.139	0.744	0.210	0.618	0.164	0.699	0.061	0.885	0.512	0.195	−0.036	0.932
Glucose (mg/dL)	−0.096	0.820	0.147	0.728	0.233	0.578	0.431	0.286	−0.123	0.772	0.125	0.768	−0.012	0.978	0.589	0.124	−0.096	0.820	0.488	0.220	0.419	0.301	−0.228	0.588
Urea (mg/dL)	0.494	0.213	0.737 *	0.037	−0.160	0.706	−0.371	0.365	0.282	0.498	0.751*	0.032	0.611	0.108	−0.123	0.772	−0.711*	0.048	−0.561	0.148	−0.731 *	0.040	−0.467	0.243
Creatine (mg/dL)	0.573	0.137	0.522	0.185	0.255	0.543	0.200	0.635	0.379	0.355	0.279	0.504	0.424	0.295	0.012	0.977	−0.488	0.220	−0.025	0.954	0.170	0.688	−0.103	0.808
Potassium (mmol/L)	0.159	0.708	−0.124	0.769	−0.304	0.464	0.333	0.420	0.578	0.134	0.177	0.674	0.170	0.688	0.304	0.464	−0.317	0.444	−0.580	0.132	−0.079	0.853	0.315	0.447
Magnesium (mg/dL)	0.212	0.614	0.000	1.000	0.117	0.782	0.542	0.165	0.185	0.661	−0.101	0.812	−0.494	0.213	0.722 *	0.043	−0.370	0.367	0.442	0.273	0.151	0.722	−0.289	0.487
Phosphorus (mg/dL)	−0.110	0.796	0.106	0.803	−0.814 *	0.014	−0.006	0.989	−0.267	0.522	−0.405	0.319	0.091	0.830	0.242	0.563	−0.360	0.381	0.148	0.726	0.236	0.573	−0.491	0.217
Serum iron (µg/dL)	0.627	0.096	−0.086	0.840	0.675	0.066	0.539	0.168	0.675	0.066	0.300	0.470	−0.275	0.509	0.356	0.387	−0.542	0.165	−0.244	0.560	−0.012	0.978	0.228	0.588
Ferritin (ng/mL)	0.554	0.154	0.160	0.706	−0.172	0.684	0.491	0.217	0.724 *	0.042	−0.100	0.814	0.216	0.608	0.184	0.662	−0.446	0.268	−0.195	0.643	0.287	0.490	0.240	0.568
Calcium (mg/dL)	0.564	0.146	−0.025	0.954	0.556	0.153	0.578	0.133	0.692	0.057	0.466	0.245	0.042	0.921	0.420	0.300	−0.648	0.082	−0.454	0.258	0.090	0.831	0.265	0.526
Hemoglobin (g/dL)	0.494	0.213	−0.049	0.908	0.552	0.156	0.695	0.056	0.638	0.089	0.375	0.359	−0.072	0.866	0.577	0.134	−0.590	0.123	−0.244	0.560	0.204	0.629	0.216	0.608
Hematocrit (%)	0.217	0.606	−0.319	0.441	0.577	0.134	0.743 *	0.035	0.491	0.217	0.275	0.509	−0.036	0.933	0.540	0.167	−0.337	0.414	−0.293	0.482	0.467	0.243	0.467	0.243
Leukocytes (10^3^/mm^3^)	−0.386	0.346	−0.209	0.620	−0.491	0.217	0.347	0.399	0.135	0.750	0.225	0.592	0.311	0.453	0.565	0.145	−0.084	0.843	−0.415	0.307	0.156	0.713	0.204	0.629
Platelets (10^3^/mm^3^)	−0.627	0.096	−0.835 **	0.010	0.209	0.620	0.287	0.490	−0.221	0.599	−0.401	0.325	−0.491	0.217	0.135	0.750	0.747 *	0.033	0.171	0.686	0.455	0.257	0.623	0.099

^‡^ Spearman Rho; ^‡‡^ bone mineral density. * *p* ≤ 0.05; ** *p* ≤ 0.001.

## Data Availability

There are restrictions on the availability of data for this trial due to the signed consent agreements around data sharing, which only allow access to external researchers for studies following the project’s purposes. Requestors wishing to access the trial data used in this study can make a request to mariscal@ugr.es.

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
