# Peer review of "Body Composition and Nutritional Status of the Spanish National Breaking Team Aspiring to the Paris 2024 Olympic Games"

_nutrients, 2023, doi:10.3390/nu15051218_

Round 1

Reviewer 1 Report

How to sure the amount of food and fruits, you did not introduce in the method. 

Line 101-101, please introduce 720 machine, country and company 

Line 103-105, how to measure bone densitometry, please provide reference and give a brief description

Please give the abbreviations of BCAA, SMM BMD in the text.

Line 189, please add reference or research link 

Line 211-217, how do you compare the function of ergogenic aid, what’s the function of ergogenic aid, in this paragraph, I don’t understand what you say.

The athlete age ranged from 18-35, in line 195-196, “This means that the bone mineral density values are optimal for the chronological age and the competitive period in which they are,” for all athlete ?  what’s the scope of femoral head T-score and a lumbar spine T-score 

Author Response

How to sure the amount of food and fruits, you did not introduce in the method.

RESPONSE: Information on the food consumption questionnaire has been introduced in the methodology section. The study population was assessed through this questionnaire by grouping foods into fruit, vegetables, protein foods and carbohydrate-rich foods, thus verifying that their intakes contained foods from all groups. The authors have included the reference on the validation of the method used on food consumption frequency.

Line 101-101, please introduce 720 machine, country and company

RESPONSE: This tool belongs to the company Microcaya@ and comes from Bilbao (Spain). This information is contained in the manuscript.

Line 103-105, how to measure bone densitometry, please provide reference and give a brief description

RESPONSE: Bone densiometry was measured by Dual energy X-ray absorptiometry (DEXA) which is the Gold-standard in research and is the method against which all other methods for similar measurements are compared. This technique uses very low doses of radiation to measure bone density and strength. Athletes signed an informed consent form for this test. It was performed without metallic objects on the body and without having performed any contrast test in the previous two weeks. The athlete is placed in a supine position, stretched out and with the arms parallel to the body without opening the fingers. An ankle strap is placed around the ankles to prevent leg movements and a check is made to ensure that the athlete is completely within the measurement frame. The test is non-invasive with no side effects and requires no anaesthesia. It takes about 6-10 minutes, depending on height and weight. The DEXA sends out an invisible thin beam with 2 energy peaks which is examined by the machine and sent to a software which is represented in a report. Pinyuan Medical@, DEXA ostodensiometer. The authors have included references.

Please give the abbreviations of BCAA, SMM BMD in the text.

RESPONSE: They are included at the foot of tables and in the text.

Line 189, please add reference or research link

RESPONSE: The authors have included the reference in the text.

Line 211-217, how do you compare the function of ergogenic aid, what’s the function of ergogenic aid, in this paragraph, I don’t understand what you say.

RESPONSE: The role of ergogenic support is to improve athletic performance through supplementation in conjunction with a varied and balanced diet. In the case of vitamin D3, this supplement was prescribed to athletes after testing them and obtaining results that were moderately deficient in this vitamin. The results show values of between 20-30 ng/dL when in the sports population they should be between 20 and 50 ng/dL, according to the series. This supplement moderately increases the range and ensures optimal intake. We have tried to clarify this in the text.

The athlete age ranged from 18-35, in line 195-196, “This means that the bone mineral density values are optimal for the chronological age and the competitive period in which they are,” for all athlete?  what’s the scope of femoral head T-score and a lumbar spine T-score

RESPONSE: The mean values of the bone mineral density test are normal for the group of breakdance athletes. The minimum value corresponds to the oldest athlete who does have a lower value in both the femoral and lumbar T-score. The data provided corresponds to the average of the group. The authors have made the corrections so that it can be clearly read that it corresponds to the mean of the study population and not to the chronological age. We are grateful for the reviewer's comment, as it was certainly important to clarify this data.

Author Response

Manuscript “Body composition and nutritional status of the Spanish National Breaking Team aspiring to the Paris 2024 Olympic Games” by Montalbán Cristina, Giménez-Blasi Nuria, García Aurora, Latorre José Antonio, Conde-Pipo Javier, López- Moro Alejandro, Mariscal-Arcas Miguel and Palacios Nieves, can be considered for publication in Nutrients after major corrections:

RESPONSE: We are grateful for the reviewer's detailed review of our manuscript and have proceeded to modify, add or delete those issues suggested by the reviewer.

Abstract: there is no word about nutritional status and methods of its establishment – remove numerical data concerning anthropometric measurements and add nutritional status data – rewrite abstract;

RESPONSE: The abstract now provides information on the methods followed in the assessment of nutritional status without anthropometric data. The study population completed questionnaires on food consumption frequency and information on ergogenic aids.

  • Line 47: what other dance specialties please name them, what about other sports: is there any comparison of Breaks to VO2max in other sports?

RESPONSE: Other dance specialities related to breaking dance can be related to Hip Hop dancers. Break dancers are a little studied group, however we have included more references as requested by the Reviewer on cardiorespiratory profile.

Line 50: dot is not placed on the end of the sentence;

RESPONSE: Corrected

  • Line 68: Olympic Games in doubled;

RESPONSE: The abbreviation "JJ" for Olympic Games has been added.

  • Line 82-83: that means” This population is representative in Spain, corresponding to 100% of the total population” 8 athletes are representative to 100% of Spanish population??? – please explain;

RESPONSE: The data refers to the population under study who are athletes selected to participate in the Olympic Games representing Spain. These 8 participants are 100% of the study population, they are not a sample of the breakdancers population, they are all those who will represent Spain in the Olympic Games. We understand that this may cause confusion and we have changed the sentence. We hope it is now clearer.

  • Methods section must be improved: issues concerning blood collection and handling prior analysis; analyzing apparatus; what is Inbody 720?; statistical description says only about spearman correlation test but authors performed also other statistical analysis: mean, SD – with what test in such a small group of participants? There is a very poor description of nutritional interview: please give more information about the questionnaire – because, as we see in the results section, there was a lot of questions concerning food products and supplements;

RESPONSE: Imbody 720 is the body compartment measurement tool, bioimpedance, used to obtain anthropometric measurements or parameters of the study population. Information on this measurement tool has already been added in the manuscript. Information on nutritional questionnaires has been added in the methodology. Information on statistical analysis has also been improved.

  • In results section for diagnostic parameters authors should include the ranges of norm for all parameters as different analyzers use different norms, the same with anthropometric parameters: various instruments using bioimpedance method give different norms for measured parameters i.e. water percentage, bone weight or fat mass; If authors were not able to perform the analysis of hormones this should be excluded from the paper and not mentioned at all.

RESPONSE: As the hormone analysis could not be carried out, this information has been removed from the document. The authors have also tried to clarify the instruments used, as well as the normal ranges of the values.

  • Line 180: authors commented results of body mass and fat percentage but not a water percentage or body muscle mass – please include this in discussion;

RESPONSE: Body water percentage is similar to other studies and muscle mass is slightly higher. This information has been included in the discussion section of the manuscript.

  • Lines 185-186: this sentence requires further analysis as cited study shows the results of participants almost 50% younger than Spanish participants possibly with other anthropometric type (mid-eastern Europe), in cited study in male break dancers body fat was lower (mean was 11% with SD 4%) and with body fat% increasing with age – I think this issue requires further discussion;

RESPONSE: The limited number of studies on this group of break dancers makes it difficult to find valid comparisons for our sample, which, in terms of age, is more heterogeneous than the studies found. We are grateful for the reviewer's comment and the authors believe that we have clarified these data in the discussion.

  • Line 191: what about data from similar type of sport performance? - for example with similar VO2max performance?

RESPONSE: In similar sports, such as theatrical dance, VO2 max was somewhat higher as break dancers have shorter exercise durations, which allows for a higher cardiorespiratory demand. We have introduced information on this question into the discussion.

  • Line:198: on what bases authors assumed that participants use non-deficient diet? It was not described in the results – as I mentioned before authors should improve paragraphs concerning nutritional interview and results of this research;

RESPONSE: We understand that the diet of our break dancers is not deficient because they complete a food consumption frequency questionnaire where the intake of food from all food groups is observed, in addition, the analyses show an optimal state of health, with a moderate deficiency in vitamin D3 that would be solved with the intake of vitamin D3 supplements. In any case, this part has been enriched with references to validation studies of the method.

  • Limitations of the study should be placed on the end of the discussion section – in this place those paragraphs blur the flow of discussion;

RESPONSE: The authors are grateful for the reviewer's comment. We have moved the limitations to the end of the discussion. It was certainly blurred.

Reviewer 3 Report

This study aims to know the characteristics of the body composition and the nutritional status of the components of the national team that aspire to participate for the first time in the Olympic Games (Olympic Games) of Paris 2024 representing Spain. The authors state that this is the first time that these characteristics have been studied in Breakers, so it is highly relevant to increase knowledge in this area in order to conduct nutritional interventions aimed at improving athletic performance in these athletes. 

I am absolutely convinced that this is an interesting subject and falls within the scope of the journal.   However, before further consideration, the following question should be well addressed by the authors.

Major issues

Is the small sample size of the survey sufficient to guarantee the accuracy of the analysis and conclusions? 

Is it proper to make no distinction between the sexes?

The data analysis strategy is not valid. In conjunction with the data analysis of LINE111-113, why was the Spearman correlation coefficient chosen to be used instead of the Kendall correlation coefficient (to differentiate between variables, such as male and female, etc.)? 

The results and discussion sections are subject to improvement.  The results are not explicitly stated and clarified. The discussion does not go far enough to highlight the main points and clarify the differences and underlying mechanisms.

Some minor issues

If the words (Introduction, objectives, results, and conclusions) are necessary in the abstract? Please confirm this based on the journal manuscript template.

The introduction section of the abstract should be reorganized.

I do not think the word “Olympic Games” is appropriate to be listed in the keywords.

Check throughout the manuscript to make sure there are no typos or grammatical errors. ​Use some cohesive words to make your article coherent.

Overall, the above issues led to the decision to reject this manuscript or to reconsider it after major revisions that well addressed these concerns.

Author Response

This study aims to know the characteristics of the body composition and the nutritional status of the components of the national team that aspire to participate for the first time in the Olympic Games (Olympic Games) of Paris 2024 representing Spain. The authors state that this is the first time that these characteristics have been studied in Breakers, so it is highly relevant to increase knowledge in this area in order to conduct nutritional interventions aimed at improving athletic performance in these athletes.

I am absolutely convinced that this is an interesting subject and falls within the scope of the journal. However, before further consideration, the following question should be well addressed by the authors.

Major issues

Is the small sample size of the survey sufficient to guarantee the accuracy of the analysis and conclusions? 

RESPONSE: The sample is composed of the only 8 members of the team that will participate in the Olympic Games. This study serves as a starting point for further studies in this type of athlete population. It seems interesting to contribute this information to the scientific literature, although we also understand that the sample is small. It has been clarified in the method that the treatment has been non-parametric. Other similar studies with break dancers also show a small sample size.

Is it proper to make no distinction between the sexes?

RESPONSE: We did not consider it appropriate to make this distinction since, of the 8 participants, only one of them is a woman. No meaningful comparisons (7 vs. 1) would be obtained in such a small sample. Moreover, at the next Olympic Games in Paris, there will be no gender distinction for this sport, so they are part of the same competition group.

The data analysis strategy is not valid. In conjunction with the data analysis of LINE111-113, why was the Spearman correlation coefficient chosen to be used instead of the Kendall correlation coefficient (to differentiate between variables, such as male and female, etc.)? 

RESPONSE: Non-parametric tests were used for the inferential treatment of the variables analysed. The authors considered the spearman correlation to be an appropriate non-parametric test. These questions have been clarified in the methodology section.

The results and discussion sections are subject to improvement.  The results are not explicitly stated and clarified. The discussion does not go far enough to highlight the main points and clarify the differences and underlying mechanisms.

RESPONSE: We believe that by providing more information on the methodology of food consumption frequency questionnaires and nutritional information, more methodological description of some of the tools used in the measurement of some parameters and a better discussion of some of the results obtained, we believe that the manuscript now presents greater overall justification of the study sample for the importance of being able to provide the scientific literature with the relevance of this research, increasing knowledge in this area and that it could serve as a starting point for future research.

Some minor issues

If the words (Introduction, objectives, results, and conclusions) are necessary in the abstract? Please confirm this based on the journal manuscript template.

RESPONSE: These words have been removed from the abstract according to the template.

The introduction section of the abstract should be reorganized.

RESPONSE: We have provided more information in the introductory part of the abstract to better justify the attractiveness and importance of this type of dance. We have also continued to reflect the fact that it is, for the first time, an Olympic sport, which is a milestone in the Olympic Games.

I do not think the word “Olympic Games” is appropriate to be listed in the keywords.

RESPONSE: This word has been removed from the keywords.

Check throughout the manuscript to make sure there are no typos or grammatical errors. Use some cohesive words to make your article coherent.

RESPONSE: The manuscript has been completely revised along these lines. We hope that it will now be syntactically and orthographically correct, and more cohesive.

Overall, the above issues led to the decision to reject this manuscript or to reconsider it after major revisions that well addressed these concerns.

RESPONSE: The authors are very grateful for the work done by the reviewers. The revision of the manuscript and the reviewers' suggestions have undoubtedly helped us to improve the manuscript, its readability and comprehensibility. We believe that it is now more compact in all its sections. Thank you very much

Round 2

Reviewer 1 Report

Accept in its current form.

Reviewer 2 Report

I have no further issues. Thank You.

Reviewer 3 Report

I fully appreciate the willingness and effort of the authors to revise the manuscript. Unfortunately, however, the revisions and explanations of the major concerns do not seem convincing to me. Also, there are still a few minor problems (such as the misusing comma instead of decimal point in the tables).  Despite this, I am personally interested in the current research and therefore encourage the authors to take more time and effort in rewriting the paper and preparing a stronger response letter.